# Time to elective surgery and its predictors after first cancellation at Debremarkos Comprehensive Specialized Hospital, Northwest Ethiopia

**Yibeltal Abiyu[1], Zewudie Aderaw[2], Lieltework Yismaw[3], Mulatu Mengaw[1], Getamesay Demelash[4], Melkamu Siferih[5]***

1 Department of Public Health, College of Medicine and Health Sciences, Debremarkos University, Debremarkos, Ethiopia, 2 Department of Public Health, Saint Paul's Hospital Millennium Medical College, Addis Ababa, Ethiopia, 3 Department of Biostatistics, College of Computational Sciences, Debremarkos University, Debremarkos, Ethiopia, 4 Department of Anesthesia, College of Medicine and Health Sciences, Debremarkos University, Debremarkos, Ethiopia, 5 Department of Obstetrics and Gynecology, School of Medicine, Debremarkos University, Debremarkos, Ethiopia

* siferihmelkamu@gmail.com

**Data Availability Statement:** Important data are available within the manuscript itself.

## Abstract

Canceling elective surgical procedures is quite common throughout Ethiopia. Despite this, there is limited evidence about the time to elective surgery after cancellation in the country. Thus, the current study aimed to determine the time to elective surgery and its predictors after the first cancellation. An institution-based retrospective follow-up study was conducted on 386 study participants at Debre Markos Comprehensive Specialized Hospital, Northwest Ethiopia, between September 1, 2017, and August 31, 2022. Utilizing a checklist, data were retrieved. To choose study participants, systematic random sampling was employed. Epi-Data version 3.1 and STATA version 14.1 were utilized. Kaplan-Meier curves and log-rank tests were employed. The Cox proportional hazard model was fitted. The mean age of the participants was 41.01 + 18.61 years. Females made up 51% of the patients. The majority were illiterate (72.3%) and resided in rural areas (70.5%). Surgery following the first cancellation had a cumulative incidence of 83.6% (95% CI: 79.6, 87.05) and an incidence rate of 32.3 per 1,000 person-days (95% CI: 29.3, 35.5). The median survival time to surgery was 25 (IQR: 17–40) days. Urban residence (AHR = 1.62; 95% CI: 1.26–1.96), being a member of health insurance schemes (AHR = 1.55; 95% CI: 1.24–1.96), stable other medical conditions (AHR = 1.43; 95% CI: 1.13–1.79), and timely completion of diagnostic tests (AHR = 1.62; 95% CI: 1.29–2.04) were significant predictors of time to surgery after first cancellation. Our study revealed that the time to surgery after the first cancellation was in the globally acceptable range and met the national target. Clinicians should focus on timely completion of diagnostic or laboratory tests, facilitating health insurance coverage, and comprehensive assessment and treatment of any coexisting medical conditions. It is urged to stratify each department's time for surgery, taking into consideration of important variables.

**Funding:** The authors received no specific funding for this work.

**Competing interests:** The authors have declared that no competing interests exist.

## Introduction

Elective surgery is defined as non-emergency surgery that can wait at least 24 hours but is medically required [1]. They are potentially life-changing, and some are same-day surgeries that do not require a hospital [2,3].

Worldwide, 3.5% of surgeries have been performed for patients who require elective surgery [1,4]. The World Health Organization (WHO) report showed that a third of elective procedures were done after patients had at least one cancellation [5,6]. After the initial cancellation, just 15% of patients in Europe and 7% of patients in Africa had to wait before surgery [7]. In Ethiopia, public specialized hospitals reported a high rate of cancellations (14.6%) [8].

Waiting times for elective surgery are a strategic complement to the quality of surgical management [8]. The majority of cut-off points are chosen fairly arbitrarily. It is necessary to establish an acceptable upper limit to ensure prompt care delivery and prevent the negative effects of waiting [9]. Whatever the type of disorder, acceptable waiting periods vary from two to twenty-five weeks [10]. Long waiting times for surgery have long been a concern in developing countries, including Ethiopia [4,11]. More than 72% of the contributing factors for long waiting times for surgery after a cancellation can be eliminated [12]. The long waiting time hinders the operating room and time from being used efficiently, causing patients and their families psychological stress [13,14].

The majority of previous time-to-surgery studies have focused on the interval of time between eligibility and surgery [4–6,11]. However, few studies have assessed the time to surgery among elective surgery cases after the first cancellation throughout the African continent, including Ethiopia; even we were unable to locate a similar study in the current study setting.

Identifying and addressing potential predictors of time to surgery can optimize scheduling, reduce patient anxiety, increase patient satisfaction, enhance overall patient outcomes, enhance the effectiveness of the healthcare system, and promote a time-to-surgery policy.

Therefore, the current study was to evaluate time to surgery and its predictors among elective surgery cases after the first cancellation at Debre Markos Comprehensive Specialized Hospital, Northwest Ethiopia.

## Methods and materials

### Ethics statement

The ethical standards conformed to the Helsinki Declaration. Ethical clearance was obtained from the Institutional Ethics Review Board (IRB) of Debremarkos University with reference number: HSC/R/C/Ser/PG/Co/50/11/14. By keeping the names of the patients anonymous, the confidentiality and privacy of the information were protected. Informed written consent was waived by the Ethics Review Board of Debremarkos University. Consent for publication was not applicable.

### Study setting, design, and participants

A five-year hospital-based retrospective follow-up study was carried out at Debre Markos Comprehensive Specialized Hospital in Northwest Ethiopia from September 1, 2017 to August 31, 2022. The data were collected from September 10 to 30, 2022.

The hospital is situated in the town of Debre Markos, which is 295 kilometers from Addis Ababa, the capital of Ethiopia, and 265 kilometers from Bahir Dar, the seat of the Amhara regional state. The hospital contains 216 beds for admission, 4 major operating rooms, 20 senior specialists who do elective surgeries, 232 staff nurses, 57 staff doctors, 10 anesthesia

professionals, 197 other health professionals, and 139 support staff members. It serves more than 3.5 million people in its catchment region and offers 24-hour service. According to the hospital's yearly records, over 1422 elective surgical procedures were performed each year, while over 280 elective procedures were canceled [15].

Source populations included all patients who were hospitalized at Debre Markos Comprehensive Specialized Hospital for elective surgery but who weren't done on the first planned day of surgery In the Departments of General Surgery, Gynecology, Obstetrics, and Orthopedics during the study period, all patients who were scheduled for elective major surgery and canceled on the first scheduled day of surgery were included in the study population. Charts without cancellation dates, the date of surgery, and charts without outcomes (surgery or censored) were excluded.

## Sample size determination and sampling technique

The sample size was determined using STATA version 14.1 (sample size analysis for Cox proportional hazards model), taking into account the hazard ratio (1.53), probability of the event (0.5), and event (174) for the variable **operation room supply** from the prior study [16]**,** as well as making assumptions about the percentage of participants who would withdraw (0.1), the two-tailed significant level of 0.05, the power of 80%, and the level of confidence (95%). The computed final sample size was 386. The patient's medical record number was extracted from the scheduled cancellation registration book. There were 1158 hospitalized patients for elective surgery who had at least one cancellation within the previous five years. The 386 study participants were chosen via proportional allocation to each year and systematic random sampling. We determined the sample interval (K) by dividing the number of units in the population by the desired sample size of each year (n = sample size of each year) (**Fig 1**)**.**

## Operational definition

Patients who refused surgery after the initial cancellation, failed to show up for the call to surgery, were waiting for an appointment at the end of the follow-up period, or died after the initial cancellation but before the appointment was censored. The event of interest was the occurrence of elective surgery after the initial cancellation. Major surgery is any procedure that puts the patient's life in danger, especially one that involves an organ like the cranium, chest, abdomen, or pelvic cavity [17]. The time to surgery was defined as the number of days from the first day of cancellation of the elective cases to the surgery date.

## Study variables

**The outcome variable** was the time to surgery after the first cancellation. **Independent variables included socio-demographic factor**s such as age, sex, place of residence, occupation, religion, marital status, educational status, and employment status; **hospital administration-related facto**rs such as availability of recovery bed, consistency of electric power supply, availability of full surgical instrument set, presence of essential anesthesia drugs, availability of OR table, oxygen supply, and presence of cross-matched blood; **patient-related factors such as a**cceptance by the patient, the patient's family or caregiver, stable other medical conditions, whether the patient was NPO, whether diagnostic or laboratory tests were completed on time, membership in community health insurance coverage, and the department in charge (obstetrics, gynecology, general surgery, and orthopedics); **health professional-related factor**s such as availability of assigned surgeons, anesthetists, and nurses in each operating room.

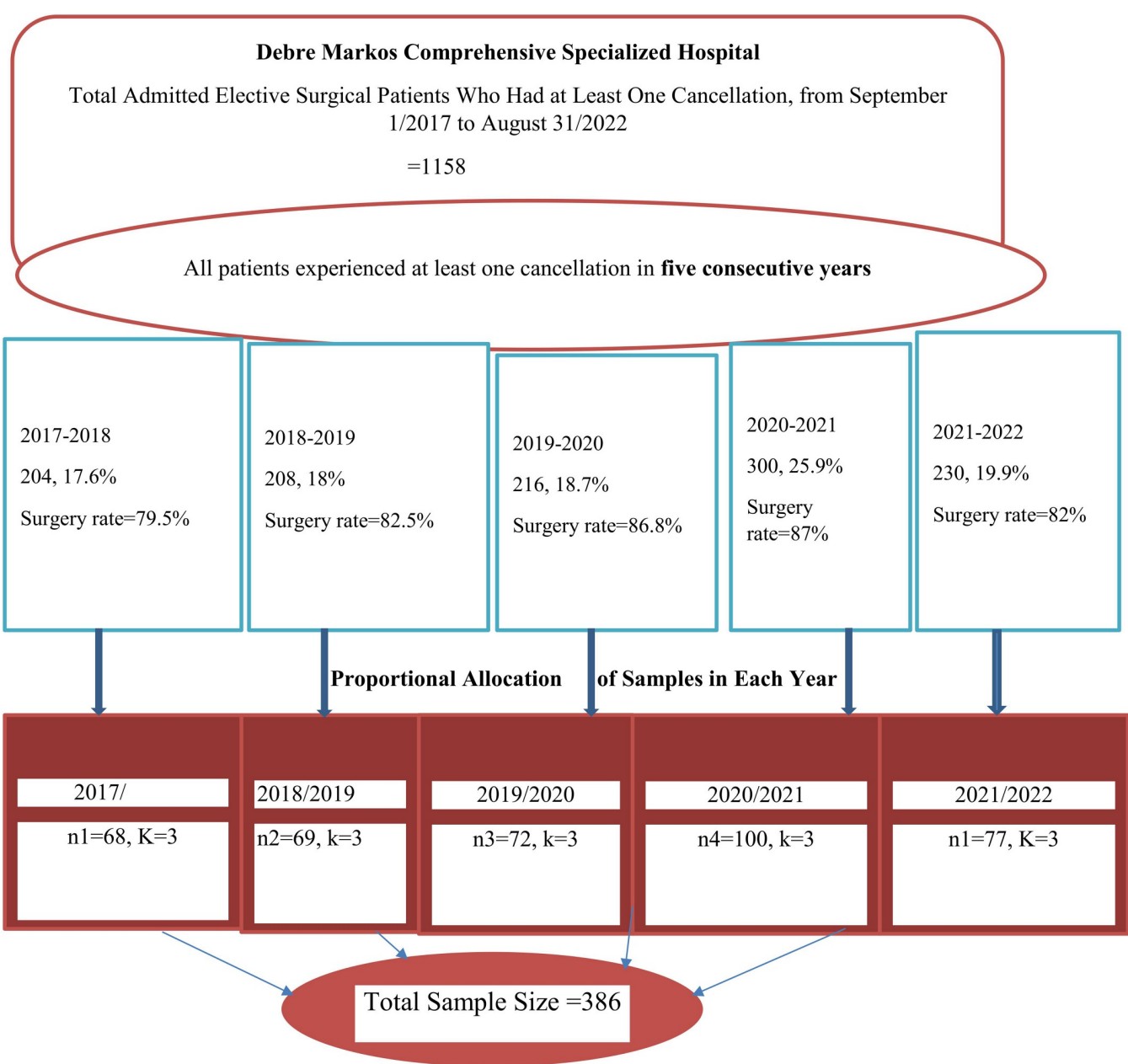

**Fig 1. Schematic presentations of the sampling procedure.**

## Data collection tools, procedures, and quality control

The data were collected using a structured checklist prepared in English. The checklist was adapted from relevant studies [7,18–23] (Fig 2) as well as from various registration books. The operation note sheet from the patient's chart was looked through to establish that surgery had been performed. In the same hospital, the tool was pretested on 5% of study participants to evaluate clarity, consistency, and understandability. Pretest data was used to modify the tool. The reliability of the tool for measuring services was 0.8 using Cronbach's Alpha. For data collection, three BSC nurses with relevant experience in the operating room were chosen. One-

**Sociodemographic Characteristics**

- Age
- Sex
- Residence
- Marital status
- Occupation
- Educational Status
- Religion
- Employment status

**Patient-related Factors**

- Being a member of community-based health insurance
- Patient acceptance of the operation
- Attendant or family acceptance
- Stable other medical conditions
- Patient kept NPO
- Ordered diagnostic or laboratory performed
- Department

**Health Professional-related Factors**

- Availability of the assigned surgeon
- Availability of the assigned anesthetist
- The presence of an operating room nurse

**Hospital Administration-related Factors**

- Availability of operation table
- Overscheduling
- Recovery bed availability
- Presence of essential anaesthesia drug
- Availability of operation room supply(gown, drape, surgical instrument set)
- Presence of cross-matched blood
- Oxygen supply

Time to Surgery after First Cancellation

**Fig 2. Conceptual framework.**

day training was given for data collectors, and close supervision and monitoring were carried out by the principal investigator. During the data collection, all data were checked for completeness and consistency.

## Data processing and analysis

Data were double entered and cleaned with Epidata version 3.1 statistical software and exported to STATA version 14.1 (College Station, Texas 77845 USA) for analysis. The data were summarized, tabulated, analyzed, and expressed with descriptive statistics such as frequency, percentage, mean, median, and interquartile range. Frequencies and percentages were presented for categorical variables, the mean for continuous variables with normally distributed data, the median, and the interquartile range for data with a non-normal distribution. Using STATA, the cumulative incidence and incidence rate were estimated. The Kaplan-Meier survival curve and log-rank test were used to estimate the survival time to surgery and to compare the survival curves, respectively. The Cox proportional hazard model was employed to identify predictors of time to surgery after the first cancellation. Variables with P≤0.25 in the bi-variate analysis were selected for the multivariable Cox proportional hazard model. The model assumptions were checked using the Schoenfeld residual test. Model fitness was assessed by the Cox-snail residual test. Before doing the multivariate analysis, the variance inflation factor (mean VIF) was used to test for multi-collinearity between the covariates. The association between variables and the time to surgery was expressed as an adjusted hazard ratio (AHR), with statistical significance set at P<0.05.

## Results

### Sociodemographic characteristics

Throughout the study period, 386 charts of patients who met the inclusion criteria were reviewed and included in the final analysis. The mean age of the participants was 41.01±18.61 years (mean±SD) and range (1–85 years). Of the patients, 51 percent were female. The majority (53.9%) were between the ages of 16 and 45, followed by those between the ages of 46 and 30 (23.8%). Most of the patients (72.3%) were illiterate and lived in rural areas (70.5%). Over half (58.3%) of patients were unemployed. Orthodox Christians constituted the vast majority of patients (94%) (**Table 1**).

**Table 1. Sociodemographic characteristics of elective surgery cases after first cancellation (n = 386).**

| Variables | Category | n (%) |
|---|---|---|
| Age | 1–15 | 27(7) |
| | 16–30 | 99(25.6) |
| | 31–45 | 109(28.2) |
| | 46–60 | 92(23.8) |
| | >60 | 59(15.3) |
| Sex | Male | 189(49) |
| | Female | 197(51) |
| Marital status | Married | 264(68.4) |
| | Single | 122(31.6) |
| Educational status | Literate | 107(27.7) |
| | Illiterate | 279(72.3) |
| Occupation | Employed | 161(41.7) |
| | Unemployed | 225(58.3) |
| Residence | Urban | 114 (29.5) |
| | Rural | 272(70.5) |
| Religion | Orthodox | 363(94) |
| | Muslim | 17(4.4) |
| | Protestant | 6(1.6) |

## Hospital administration-related, patient-related, and professional-related factors

The majority (82.66%) of patients did not have an electricity interruption during the preoperative period; 62.85% of them had a recovery bed available; and 66.6% of them had essential anesthesia drugs present. But more than half (57.9%) lacked access to cross-matched blood. There was no extra scheduling by the liaison office for 76.2% of patients. Nearly half (54.15%) of the study participants were enrolled in community health insurance schemes. The majority of patients (87.31%) and family members or attendants (85.49%) agreed to the procedure. Preoperative diagnostic or laboratory tests were carried out on 58% of all patients before surgery. The majority of participants (72.3%) had a stable other medical condition. The majority (39.6%) of elective major cases that were canceled came from the general surgery department, followed by gynecology (25.9%), orthopedics (18.7%), and obstetrics (15.7%) in that order. Assigned surgeons, operating room nurses, and anesthetists were present for 71.8%, 92.5%, and 77.2% of patients at the time of operation, respectively (**Table 2**).

## The incidence and Kaplan-Meier survival estimates of elective surgery after the first cancellation

A total of 323 patients underwent surgery, and 63 patients were censored, yielding a cumulative incidence of surgery of 83.6% (95% CI: 79.6, 87.05) over the follow-up period. Out of the patients that were censored, 50% had not undergone surgery at the end of the follow-up period, 26.65% had been sent to other hospitals, 18.75% did not undergo surgery owing to unstable medical conditions, and 7.81% died before the procedure was performed. The total follow-up time was 10013 person-days with an incidence rate of 32.3 per 1,000 person-day observations (95% CI 29.3–35.5). Elective surgery after cancellation could be done in as little as one day or as long as ninety days. The overall Kaplan-Meier estimate showed that the probability of surgery following the first cancellation was high on the first few days after the first cancellation and fell as the follow-up time increased. The median survival time for surgery was 25 days, with an IQR of 17–40 days. The mean survival time to surgery was 30.62 (95% CI 28.56–32.67) days (**Fig 3**). The Kaplan-Meier curve with log-rank p-value showed significant differences in the estimate of time to surgery between the categories of variables: place of residence, timely completed diagnostic or laboratory tests, community health insurance membership, and status of other medical condition (**Fig 4**).

## Predictors of time to surgery

The final Cox proportional hazard model revealed that urban residence, stable other medical conditions, timely ordering and completion of laboratory testing, and health insurance membership were significant predictors of time to surgery at p-value< 0.05. The variance inflation factor (mean VIF) used to test multi-collinearity was checked before the multivariable Cox proportional hazard model, and the result was 1.12, indicating the nonexistence of multicollinearity between covariates. According to the Schoenfeld residuals' global test result, all of the covariates met the proportional hazard assumption (chi-square = 14.56 and p-value = 0.78). The overall model fitness of the data in the Cox proportional hazards regression model was demonstrated by the Cox-Snell residual and Nelson-Alen cumulative hazards graphs. The hazard function followed the 45-degree line very closely, indicating that the model fit the data well (**Fig 5**).

Keeping other variables constant at a given point in time, the probability of a shorter time to surgery among urban residents was 1.62 times higher as compared to rural residents

**Table 2. Hospital administration-related, patient-related, and professional-related factors.**

| Hospital administration-related factors | Category | n (%) |
|---|---|---|
| The availability of a recovery bed | Yes | 203(62.85) |
| | No | 120 (37.15) |
| No interruption of the electrical supply | Yes | 267 (82.66) |
| | No | 56 (17.34) |
| Essential anesthesia drugs were present | Yes | 215(66.56) |
| | No | 108(33.44) |
| Cross-matched blood was available. | Yes | 136(42.11) |
| | No | 187(57.89) |
| Accessibility of an oxygen supply | Yes | 277(85.76) |
| | No | 46(14.24) |
| Gown and drapes were present | Yes | 239(73.99) |
| | No | 84(26.01) |
| A complete set of surgical instruments was present | Yes | 191(59.13) |
| | No | 132(40.87) |
| The schedule was too much | Yes | 77(23.84) |
| | No | 246(76.16) |
| An operating table was available | Yes | 259 (80.19) |
| | No | 64(19.81) |
| **Patient-related factors** | | |
| Enrolled in community health insurance | Yes | 216(56) |
| | No | 170 (44.04) |
| Acceptance of the operation by the patient | Yes | 337(87.31) |
| | No | 49 (12.69) |
| The family's agreement to the operation | Yes | 330(85.49) |
| | No | 56(14.51) |
| The patient remained NPO | Yes | 306 (79.27) |
| | No | 80 (20.73) |
| Stable other medical conditions | Yes | 279(72.28) |
| | No | 107 (27.72) |
| Diagnostic or laboratory tests completed timely | Yes | 224 (58.03) |
| | No | 162(41.90) |
| Department | General Surgery | 153(39.6) |
| | Gynecology | 100(25.9) |
| | Obstetrics | 61(15.8) |
| | Orthopedics | 72(18.7) |
| **Professional-related factors** | | |
| The assigned surgeon was available. | Yes | 277(71.8) |
| | No | 109(28.2) |
| Availability of assigned anesthetist | Yes | 298(77.2) |
| | No | 88(22.8) |
| The presence of an operating room nurse | Yes | 357(92.5) |
| | No | 29(7.5) |

(AHR = 1.62; 95% CI 1.26, 1.96). Similarly, patients whose diagnostic or laboratory tests were timely requested and completed had 62% (AHR = 1.62; 95% CI 1.29, 2.04) faster time to elective surgery than patients whose tests were not completed, by adjusting for other confounders. Moreover, patients who were enrolled in a health insurance scheme had a 55% shorter time to

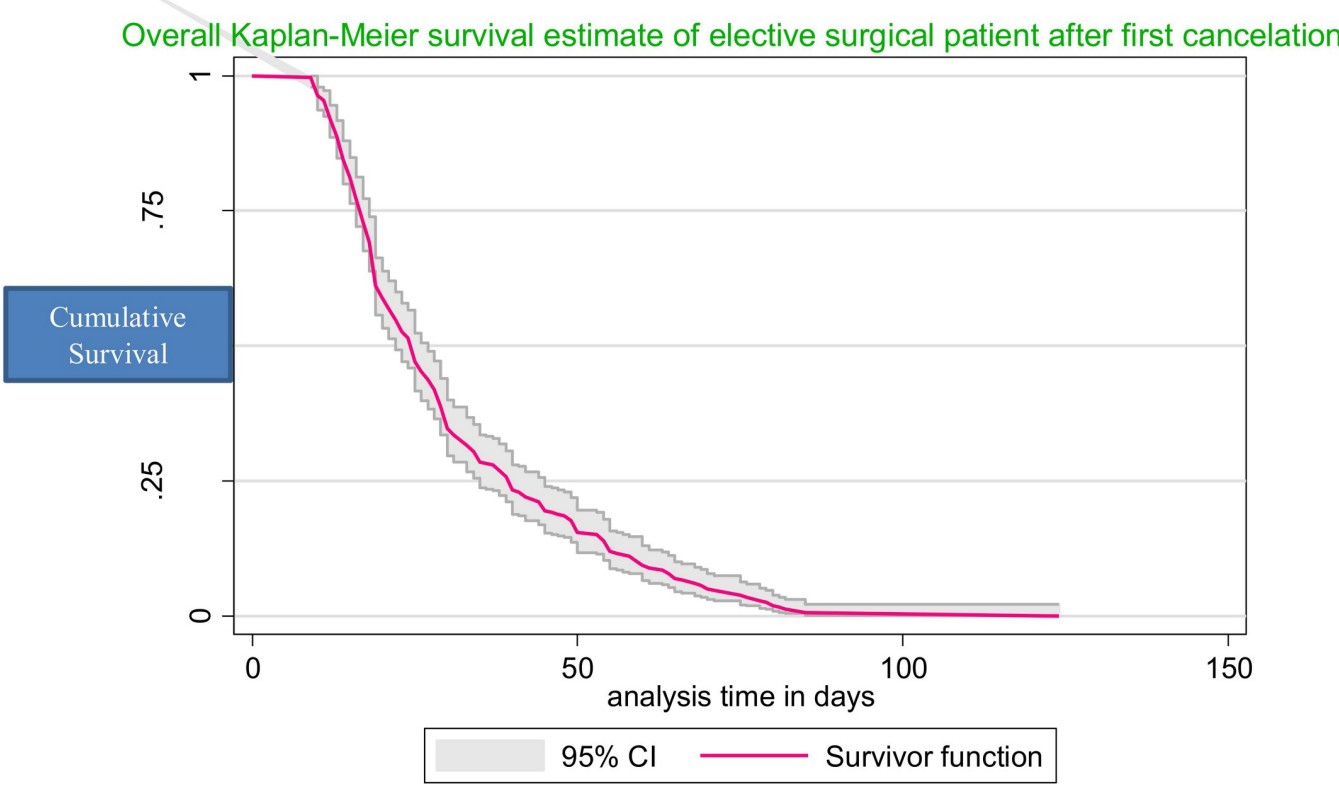

**Fig 3. Overall Kaplan-Meier survival estimate of time to elective surgery after first cancellation.**

surgery following first cancellation than those with no health insurance coverage (AHR = 1.55, 95% CI 1.24, 1.96). Finally, holding other variables constant, patients with stable other medical conditions had a 43% (95% CI: 1.13–1.79; AHR = 1.43) faster time to surgery than patients with unstable other medical conditions (**Table 3**).

## Discussion

This study was conducted to determine the time to elective surgery and its predictors after the first cancellation at Debremarkos Comprehensive Specialized Hospital, Northwest Ethiopia. The cumulative incidence of surgery following the initial cancellations was 83.6%. The median survival time for elective surgery was 25 days, and the mean survival time for elective surgery was 30.62 days. Residence, community health insurance membership, timely completed diagnostic tests, and status of other medical conditions were independent predictors of time to elective surgery.

Our study showed that after the initial cancellations, the cumulative incidence of elective surgery was roughly comparable to a study conducted at Zewditu Memorial Hospital in Ethiopia (86%) [12]. Likewise, it was similar to studies conducted in Uganda (80%), the UK (80%), and Iran (87%) [1,24,25]. However, the findings are lower than those of the studies from Canada (92%), Burkina Faso (89%), and India (95%), which were published in studies [19,26,27]. The quality of operating rooms, sufficient staff training, level of public knowledge, modern equipment, and high availability of services in those nations may all contribute to a greater rate of elective surgery. On the other hand, compared to studies conducted in Zambia (77%) [24], Ethiopia (70.8%) [28], and Ghana (50%) [29], the incidence in our study is higher. The

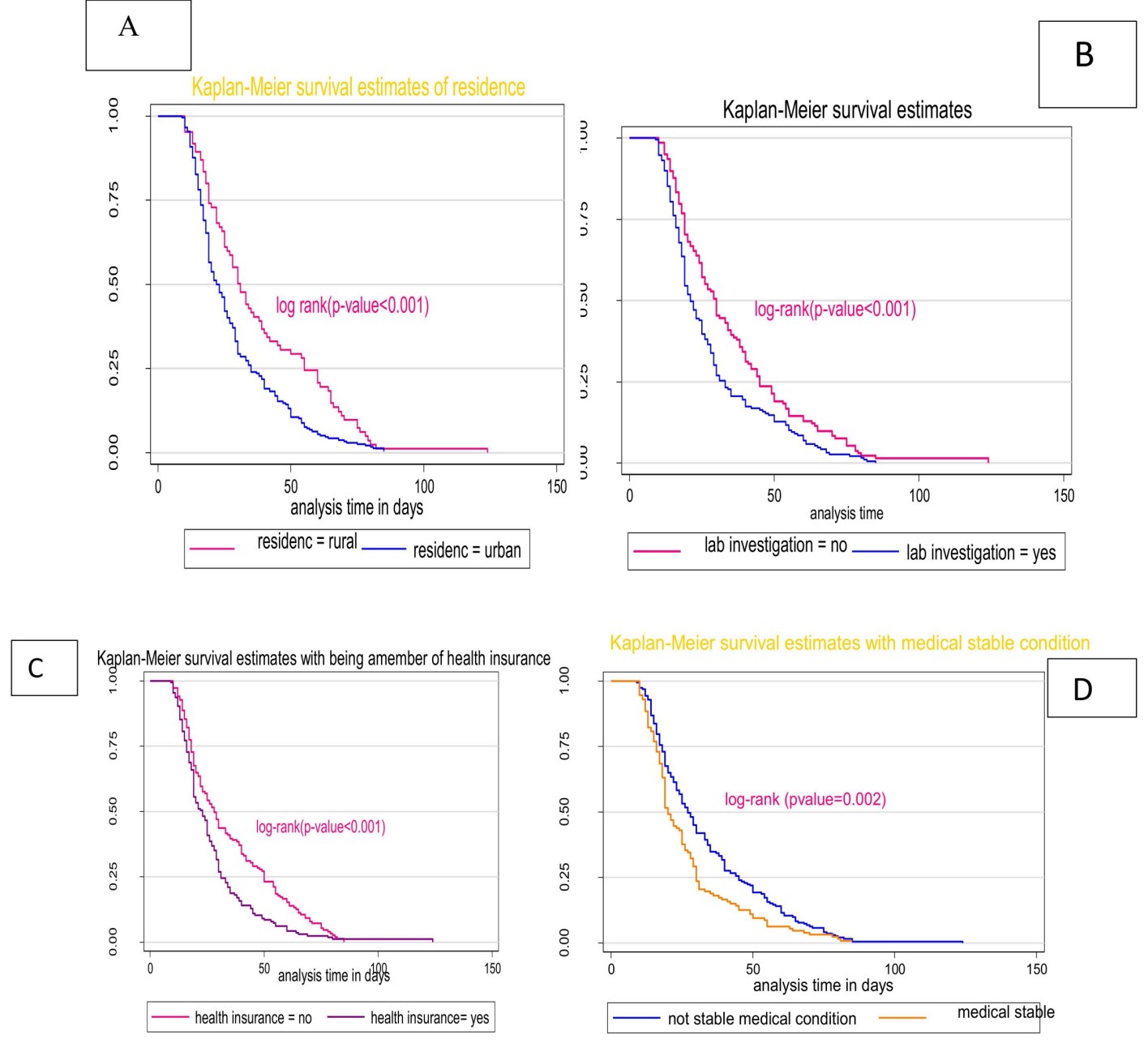

**Fig 4.** Kaplan-Meier survival estimates for categorical variables, A. for residence, B. for diagnostic or laboratory tests done, C. health insurance membership, D. for status of medical condition.

discrepancy may be accounted for by the disparities in the study population, design, duration, and catchment area low patient flow.

The mean survival time to elective surgery in the current study was equivalent to the 30-day projected time to elective surgery by the most recent Saving Lives through Safe Surgery Plan (SaLTS II) of the Ethiopian Federal Ministry of Health (FMOH) [30]. This new approach describes all elective surgery cases rather than mentioning specific patients who had their procedures canceled. The result is also comparable to the recent 36-day average wait time for surgery in the country [8]. Both the median and mean time to elective surgery following the first

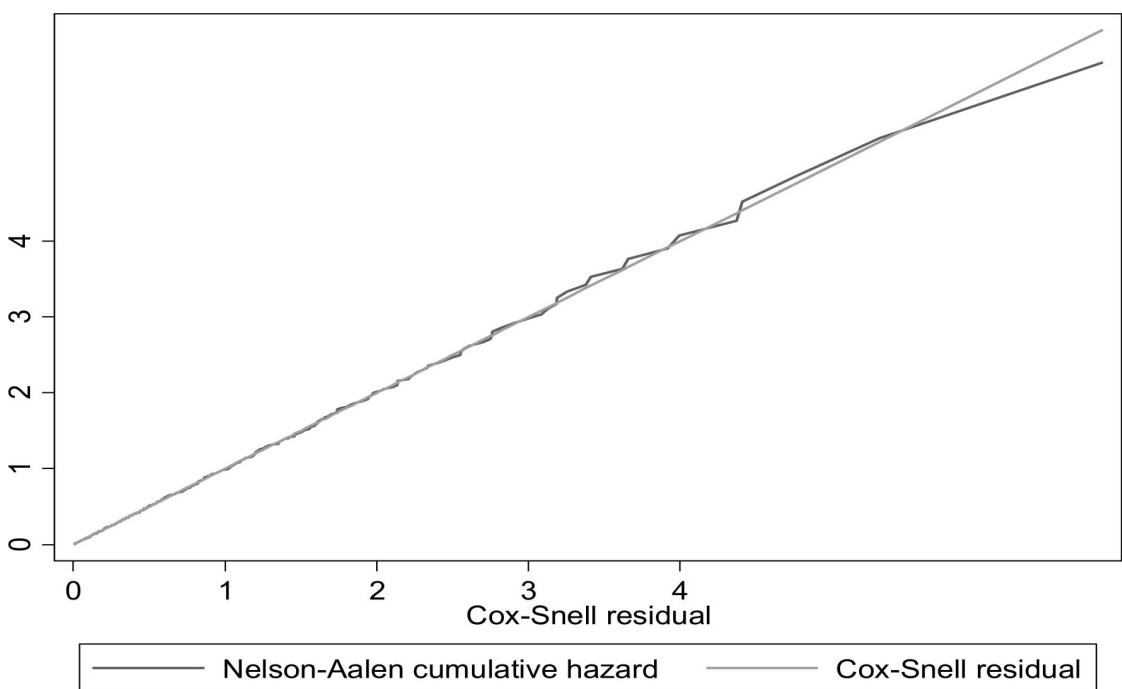

**Fig 5. Cox-Snell residual Nelson-Alen cumulative hazard graph for time to elective surgery.**

cancellation are within the most acceptable global range (2 to 25 weeks) [10]. Also comparable to 40 days in Australia was the median time to elective surgery [13]. According to the narrative review [31], the current finding is shorter than the average waiting times of 3 to 6 months in the United Kingdom, Sweden, and New Zealand, and 1.5 months in the Netherlands. The fact that we only studied those patients who had the initial cancellation could help to explain this.

As compared to rural residents, urban residents had a 62% faster time to elective surgery after the first cancellation. The findings are in harmony with those of previous studies conducted in Tanzania, Portugal, and Ethiopia [16,32–34]. Similar to urban residents, patients who promptly performed their diagnostic or laboratory tests following the initial cancellation had a 62% reduced wait time to surgery compared to patients who didn't. This is consistent with earlier studies in Wales [35] and Nigeria [36]. Moreover, those who were enrolled in community health insurance coverage schemes had a 55 percent shorter wait time for elective surgery than those who were not. This finding is supported by studies conducted in Australia [37] **and** Nigeria [36]. Because they paid per family per year at a lower cost and were, therefore, more likely to visit public hospitals, these individuals had a faster time to surgery after their first cancellation. Finally, another factor that predicted the time to surgery after the first cancellation and shortened it by 43% was the stable state of health before elective surgery. This study coincides with a study in Ethiopia's Hawasa Comprehensive Specialized Hospital [38], as well as one conducted in Portuguese [16]. It is reasonable to assert that a stable other medical condition is necessary before elective surgery because it is challenging to induce anesthesia, necessitates rescheduling, and extends the time required for the procedure itself.

It was a strength that the five years were deemed adequate for study. Concerning the time to surgery and its predictors following the initial cancellation, our study provided the first data in the country. The study employed a good model that produces a more accurate estimate of the time to surgery and its predictors. The study had limitations due to its retrospective nature. Data collectors possibly included cases with complete data, which was taken as the main

**Table 3. Predictors of time to elective surgery after the first cancellation (n = 386).**

| Variables | Category | Surgery | | CHR((95%CI) | AHR(95%CI) |
|---|---|---|---|---|---|
| | | Event | Censored | | |
| Sex | Male | 162 | 32 | 1 | 1 |
| | Female | 161 | 31 | 0.71[0.56–0.88] | 0.74[0.59–0.93] |
| Occupation | Employed | 144 | 27 | 1 | 1 |
| | Unemployed | 189 | 36 | 0.70 [0.56–0.88] | 0.76[0.60–0.96] |
| Residence | Urban | 83 | 17 | 1.58[1.22–2.04] | 1.62[1.26–2.09]** |
| | Rural | 240 | 46 | 1 | 1 |
| Educational status | Literate | 177 | 20 | 1.20[0.95–1.51] | 1.19[0.95–1.50] |
| | Illiterate | 206 | 43 | 1 | 1 |
| Availability of assigned surgeon | Yes | 220 | 45 | 1 | 1 |
| | No | 103 | 18 | 1.27[1.00–1.61] | 1.33[1.05–1.70] |
| Community health insurance coverage | Yes | 172 | 39 | 1.48[1.18–1.85] | 1.55[1.24–1.96]** |
| | No | 151 | 24 | 1 | 1 |
| Stable other medical conditions | Yes | 195 | 40 | 1.39[1.11–1.74] | 1.43[1.13–1.77]* |
| | No | 128 | 23 | 1 | 1 |
| Timely completion of diagnostic or laboratory tests. | Yes | 189 | 28 | 1.44[1.15–1.80] | 1.62[1.29–2.04]** |
| | No | 134 | 35 | 1 | 0.83[0.66–1.05] |
| Full surgical instrument set | Yes | 191 | 52 | 1 | 1 |
| | No | 132 | 11 | 1.34[1.07–1.67] | 1.18[0.94–1.49] |
| Presence of an oxygen supply | Yes | 269 | 58 | 1 | 1 |
| | No | 54 | 5 | 1.21[0.90–1.63] | 1.06[0.78–1.43] |
| Cross-matched blood was available | Yes | 136 | 24 | 1 | 1 |
| | No | 187 | 39 | 0.85[0.68–1.06] | 0.86[0.68–1.09] |
| The availability of a recovery bed | Yes | 213 | 50 | 1 | 1 |
| | No | 110 | 13 | 1.17[0.93–1.48] | 1.12[0.89–1.42] |

* P-value<0.05

** P-value<0.001.

drawback of the study. Other unmeasured variables, such as socioeconomic status, the distance from the place of residence, psychological variables, and the type of operation to be performed, may affect the findings. The comparison was challenging due to the dearth of similar studies, particularly for the study population (elective cases after cancellation) and the primary outcome of interest (median survival time). Instead of focusing on the median survival time to elective surgery, several prior studies attempted to explain the average waiting times. Because all main departments share the same surgical rooms, the inclusion of obstetric patients was another limitation that might impact our findings. Our study could not address the waiting times for surgical procedures in each department. Given the limitations listed above, the findings of this study should be rated with caution.

## Conclusion and recommendation

This retrospective follow-up study evaluated the time to surgery after the first cancellation and identified its predictors using the Cox proportional hazard model. The time to surgery was in the globally acceptable range and met the national target. Community health insurance members, urban residents, patients with timely completed diagnostic or laboratory tests, and those with stable other medical conditions had a shorter wait for elective surgery. Clinicians should focus on prompt completion of diagnostic or laboratory tests, promoting health insurance

schemes to patients, and addressing concomitant medical conditions. Further studies should focus on stratifying the time to surgery for each department using multicenter prospective cohort studies that incorporate important factors.

## Acknowledgments

We are grateful to Dube Jara for his prudent advice and compassionate support.

## Author Contributions

**Conceptualization:** Yibeltal Abiyu, Zewudie Aderaw, Lieltework Yismaw, Mulatu Mengaw, Getamesay Demelash.

**Data curation:** Yibeltal Abiyu, Zewudie Aderaw, Melkamu Siferih.

**Formal analysis:** Zewudie Aderaw, Lieltework Yismaw, Mulatu Mengaw, Getamesay Demelash, Melkamu Siferih.

**Funding acquisition:** Melkamu Siferih.

**Investigation:** Yibeltal Abiyu, Zewudie Aderaw, Lieltework Yismaw, Mulatu Mengaw, Getamesay Demelash, Melkamu Siferih.

**Methodology:** Yibeltal Abiyu, Zewudie Aderaw, Lieltework Yismaw, Mulatu Mengaw, Getamesay Demelash, Melkamu Siferih.

**Project administration:** Yibeltal Abiyu, Zewudie Aderaw, Lieltework Yismaw, Mulatu Mengaw, Getamesay Demelash, Melkamu Siferih.

**Resources:** Yibeltal Abiyu, Zewudie Aderaw, Lieltework Yismaw, Mulatu Mengaw, Getamesay Demelash, Melkamu Siferih.

**Software:** Yibeltal Abiyu, Zewudie Aderaw, Lieltework Yismaw, Mulatu Mengaw, Getamesay Demelash, Melkamu Siferih.

**Supervision:** Yibeltal Abiyu, Zewudie Aderaw, Lieltework Yismaw, Mulatu Mengaw, Getamesay Demelash, Melkamu Siferih.

**Validation:** Yibeltal Abiyu, Zewudie Aderaw, Lieltework Yismaw, Mulatu Mengaw, Getamesay Demelash, Melkamu Siferih.

**Visualization:** Yibeltal Abiyu, Zewudie Aderaw, Lieltework Yismaw, Mulatu Mengaw, Getamesay Demelash, Melkamu Siferih.

**Writing – original draft:** Yibeltal Abiyu, Zewudie Aderaw, Lieltework Yismaw, Mulatu Mengaw, Getamesay Demelash, Melkamu Siferih.

**Writing – review & editing:** Yibeltal Abiyu, Zewudie Aderaw, Lieltework Yismaw, Mulatu Mengaw, Getamesay Demelash, Melkamu Siferih.

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
