## [Decision Letter · Decision Letter 0]

25 Jul 2023

PGPH-D-23-00985

Time to Surgery after First Cancellation and Its Predictors in Elective Surgery Cases at Debremarkos Comprehensive Specialized Hospital, Northwestern Ethiopia: Time to Event Analysis

Dear Dr. Zeleke,

Thank you for submitting your manuscript to PLOS Global Public Health. After careful consideration, we feel that it has merit but does not fully meet PLOS Global Public Health’s publication criteria as it currently stands. Therefore, we invite you to submit a revised version of the manuscript that addresses the points raised during the review process.

Thank you for submitting this manuscript to the journal. I found the manuscript quite intriguing as it delves into a critical matter—the timing of surgery after the initial cancellation and the factors influencing it in elective procedures. These elements carry substantial implications for both patient outcomes and the efficiency of healthcare systems, and can help with the fine-tuning of scheduling, alleviation of patient anxiety, and result in improved patient satisfaction and the overall delivery of surgical care.

Please apply all changes identified and suggested by reviewers for acceptance, as the article needs a very major revision to be acceptable.

There is a consensus between reviewers and the editorial team that the manuscript is not written in clear grammatical style, and the text is difficult to understand. PLOS Global Public Health does not copyedit accepted manuscripts, so the language in submitted articles must be clear, correct, and unambiguous. Consider using a word processing editor (like *Grammarly*) to address this. Passing this through native or L2 English speakers with expertise in grammar would also be helpful to significantly improve this draft.

Please address the reviewer's concern on the veracity of your methods/data adequately. Clarify your sample size calculation. Your discussion needs much more clarification and  the study conclusions should move beyond recounting of results to an effective conclusion of the study. Recommendations should be based only on the findings of the study. Please ensure a complete and thorough revision of the entire work.

Please note that this decision is justified on PLOS Global Public Health’s publication criteria and not a decision based on novelty or perceived impact.

We look forward to receiving your revised manuscript.

Kind regards,

Barnabas Tobi Alayande

Academic Editor

Journal Requirements:

Additional Editor Comments (if provided):

Reviewers' comments:

Reviewer's Responses to Questions

**Comments to the Author**

1. Does this manuscript meet PLOS Global Public Health’s publication criteria? Is the manuscript technically sound, and do the data support the conclusions? The manuscript must describe methodologically and ethically rigorous research with conclusions that are appropriately drawn based on the data presented.

Reviewer #1: No

Reviewer #2: Yes

Reviewer #3: Yes

2. Has the statistical analysis been performed appropriately and rigorously?

Reviewer #1: Yes

Reviewer #2: Yes

Reviewer #3: Yes

3. Have the authors made all data underlying the findings in their manuscript fully available (please refer to the Data Availability Statement at the start of the manuscript PDF file)?

Reviewer #1: Yes

Reviewer #2: Yes

Reviewer #3: Yes

4. Is the manuscript presented in an intelligible fashion and written in standard English?

Reviewer #1: No

Reviewer #2: No

Reviewer #3: No

5. Review Comments to the Author

Reviewer #1: I read the manuscript with interest. It addresses a very important issue: the timing of surgery following the first cancellation and its predictors in elective procedures. These factors can have a significant impact on patient outcomes and healthcare system efficiency. By identifying and addressing potential predictors, we can optimize scheduling, reduce patient anxiety, enhance patient satisfaction, and improve overall surgical care delivery.

I herewith forward my input

Abstract

1. The abstract result section lacks some more data rather they gave detail conclusion content some of which are outside their finding.

Introduction

2. On page 3, the way the text is written looks like result or discussion section.

3. ‘’ Our observations showed that many patients suffer from severe pain, additional costs, emotional trauma, and feelings of hopelessness, and eventually, the patient may die if the operation is canceled again on the scheduled date.’’

4. On same page just after the above paragraph, the sentence lacks clarity and it would be good to rewrite as

a. ‘’ A better assessment and understanding of factors that affect time to surgery after a first due date cancellation can help to reduce complications, re-operations, and unnecessary hospital stays. In turn, this can improve quality of care for patients who have undergone elective surgery.’’

5. in at Debre Markos Comprehensive Specialized Hospital, Northwestern Ethiopia.

6. The following sentence need rewriting.

''This study will help clinicians and other service providers design interventions to reduce surgical time in hospitals by removing predictable factors that could be avoided.''

Method

7. Sample size calculation (Fig 1): how they were sure to achieve the target sample size and did not have contingency?

8. It is good authors made sampling proportionally allocated for each year. However, why they could not do that for each department (General Surgery, Gynecology, Obstetrics, and Orthopedics)? As the authors had not made a proportional allocation of the departments, how did they address such confounding by indication factors? Another sticking issue, how did they deal with Gyn/Obs cases? What if a scheduled CS delivery cancelled for obvious reasons (initially schedule could because of risk to the mother and latter risk could be ruled out).

9. Operational definition: This section is not should be placed separately. These terms should be explained under each variable. E.g state the outcome variable and define it, the same should apply for other operation definitions to be placed under each respective.

a. Authors can also those definitions as supplement file (if need be).

10. Overall, the method lacks clarity how data was collectors for non-patient related factors. How could it be possible to get such detail information for surgery that was cancelled 4-5 years ago? (For instance, adequate OR table (Y/N: where did they get this information? I am not sure if this data is documented on patient chart. Table 5 data looks unrealistic all add up 386 (was this too from patient chart?)

11. Table 6 also indicates no missing case at all. How did the authors achieve this precision? They might have selected patient cases only with 100% complete data with risk of selection bias.

Result

12. In all tables put Frequency and % in one column as N (%)

13. For marital status I would suggest making it Married /Single regardless of the reason for being single. That give good number for analysis and interpretation.

14. Put Table 1 and 2 as Table 1, and important to regroup logically (eg biological children adult and geriatric).

15. I would suggest putting Table 3, 4 and 5 as one Table 2 segregated by the three factors.

16. Refer my #10 feedback on method section.

Reviewer #2: Your title is very interesting and it is important to guide the clinical practice. But, I have the following concerns to be addressed by the authors.

Abstract

1. The total person-days of observation were 10013. What does it mean?

2. The in-patient surgery rate after the first cancellation was 83.6% despite many patients requiring surgery, but comparable to many previous findings. It doesn't give a meaning. Please re-write it again

3. Your recommendation should't be obligatory and based on the stud findings. So, try to modify it

Introduction

4. In paragraph one and two, there is a redundancy of ideas. So, better to write in one sentence since the two sentences have the same idea.

5. In paragraph 3, "Previously, time-to-surgery among select cases on the African continent, particularly Ethiopia,

focused primarily on time from eligibility to surgery". Re-write it again

Methods

6. The description of your study setting is full of language and editorial errors that needs major modification

7.Why you exclude minor surgeries?

8. In sample size determination, "STATA command 'power log-rank (cumulative probability of survival), HR (1.53) power (0.8) wd prob (0.1 (15) giving same sample size." It is not clear. What was the probability of the event? and

What was the final sample size based on the STATA command?

9. Why you allocated the sample size proportionally to each year?

10. Why you preferred systematic random sampling over SRS if you assumed the population is homogeneous and had a sampling frame?

11. These operational definitions are not clear for readers and scholars. So, try to make them measurable to answer the questions; when we can say there is adequate number of surgeons, nurses...

12. What if patients who were waiting the appointment at the end of your follow-up period? Did you have a right censored? If no, come with your strong justifications.

13. What is the difference between survival time and time to surgery in your study? Which one is your primary objective? How did you calculate it?

14. How did you check the PH assumptions? What was the result of global test? Which graph did you use to check the PH assumptions?

15. In data processing and analysis; " Likely hood ratio (LR) was used to identify model fitness among the candidate's

model and the model with high value considered well fit indicating less information lost on the data was selected."

Did you perform a model comparison? If so, how did you select Cox from other candidate models?

16. Better to put ethical consideration in the declaration section

Result

17. Using two or more measures of central tendency and/or dispersion at the same time is not appropriate.

18. Try to cite and enter each table at the end if each paragraph

19. Table 2 has too many categories, better to include age in socio-demographic variables and try to minimize the number of categories.

20. You have to present some information in the text form after each subsection of your results and then cite the table for detail

21. In table 3, all are not variables instead they are factors. So, you need to understand the difference between variables and factors. How did you score these variables for each patient? These are institutional resource-related variable and it is difficult to score for each patient. I need a strong justification. The same is true for table 4 and 5.

22. "The overall Kaplan-Meier estimate showed that the probability of surgery of elective case surgery

was a long duration on the first day after the first cancellation and progressively short time as the

follow-up time increased as shown in the figure below" What does it mean? How did you interpret the KM survival curve? What kind of information did you get from this curve?

23. Why you summerize the survival time using both median and mean? I think median is better and enough too.

Discussion

24. Your discussion section need a major revision

Conclusion and recommendations

25. You failed to conclude you findings, simply repetition of the result section. Please try to revise accordingly.

26. Recommendations should be based on the findings of the study and they should be amenable, targeted and action oriented. Try to modify based on your findings.

Reviewer #3: . The study explored the time to surgery and its predictors among elective surgery cases after the first cancellation in Debre Markos Comprehensive Specialized Hospital. The findings of the study will help improve our understanding of time to surgery and promote policies regarding time to surgery. However, there are some comments which need to be considered in order to improve the quality of the manuscript. I suggest that the authors should revise the manuscript, taking into consideration the following comments and suggestions:

1. To allow for easy review and referencing in specific sections of the manuscript, it is advisable to set line numbers.

2. There are grammatical and punctuation errors which need to be corrected in the manuscript.

3. What does WHO mean in the introduction section? Kindly give the full meaning of any abbreviation at the first mention.

4. Tables 1 and 2 can be merged into one table instead of being presented separately.

5. Some tables should be conformed to the scientific style of formatting tables.

6. The second sentence of the first paragraph of the Time to Surgery in Elective Surgical Patients After the First Cancellation section needs to be rephrased.

7. Under the discussion section, in paragraph four on page 19, one of the studies used for comparison was stated as “study in a specialized hospital in Ethiopia” needs to be rephrased since it lacks clarity, and which specialized hospital findings did you want to talk about and compare your study with?

8. Figure numbers in the text and under the, "supplementary materials section are inconsistent. Better to re-number it.

9. There is inconsistency in the reference list too.

6. PLOS authors have the option to publish the peer review history of their article (what does this mean?). If published, this will include your full peer review and any attached files.

**Do you want your identity to be public for this peer review?** For information about this choice, including consent withdrawal, please see our Privacy Policy.

Reviewer #1: No

Reviewer #2: No

Reviewer #3: No

---

## [Decision Letter · Decision Letter 1]

24 Nov 2023

PGPH-D-23-00985R1

Time to Elective Surgery and Its Predictors after First Cancellation at Debremarkos Comprehensive Specialized Hospital, Northwest Ethiopia

Dear Dr. Zeleke,

Thank you for submitting your manuscript to PLOS Global Public Health. After careful consideration, we feel that it has merit but does not fully meet PLOS Global Public Health’s publication criteria as it currently stands. Therefore, we invite you to submit a revised version of the manuscript that addresses the points raised during the review process.

1. The manuscript has been significantly improved from its last iteration as evidenced by the reviews of both reviewers. A lot of work has been put into clarifying and strengthening the message of the paper, and this is strongly acknowledged. However one reviewer has raised concerns that need to be addressed before the paper can be accepted for publication.

Please edit the abstract in response to the reviewer's comments (see attachment).

2. To address the concern of the reviewer in the methods, the authors need to clarify the variables collected in the surgery cancellation register and let the readers know if the variables in contention are routinely collected at at Debremarkos Comprehensive Specialized Hospital. Further clarity would be enhanced if an additional appendix (supplementary material) with a list of all variables routinely collected in the surgery cancellation register at the hospital is included.

3. The authors need to also justify the perfect data without missingness in order to address the concern by the reviewer of possible *selection bias* by including only cases with 100% complete data. See comments 5 and 6 in the attachment.

4. Also address comment 7 via a minor edit.

All these comments are addressable, and we would like you to focus on these minor points in your review. Hopefully, with one more brief round of pointed review, the manuscript should be in proper shape to disseminate widely as this is a very insightful paper. Congratulations to the authors for a commendable job.

We look forward to receiving your revised manuscript.

Kind regards,

Barnabas Tobi Alayande

Academic Editor

Journal Requirements:

Additional Editor Comments (if provided):

Reviewers' comments:

Reviewer's Responses to Questions

**Comments to the Author**

1. If the authors have adequately addressed your comments raised in a previous round of review and you feel that this manuscript is now acceptable for publication, you may indicate that here to bypass the “Comments to the Author” section, enter your conflict of interest statement in the “Confidential to Editor” section, and submit your "Accept" recommendation.

Reviewer #1: (No Response)

Reviewer #2: All comments have been addressed

2. Does this manuscript meet PLOS Global Public Health’s publication criteria? Is the manuscript technically sound, and do the data support the conclusions? The manuscript must describe methodologically and ethically rigorous research with conclusions that are appropriately drawn based on the data presented.

Reviewer #1: Partly

Reviewer #2: Yes

3. Has the statistical analysis been performed appropriately and rigorously?

Reviewer #1: I don't know

Reviewer #2: Yes

4. Have the authors made all data underlying the findings in their manuscript fully available (please refer to the Data Availability Statement at the start of the manuscript PDF file)?

Reviewer #1: Yes

Reviewer #2: Yes

5. Is the manuscript presented in an intelligible fashion and written in standard English?

Reviewer #1: No

Reviewer #2: Yes

6. Review Comments to the Author

Reviewer #1: For detail, see attachment.

Reviewer #2: (No Response)

7. PLOS authors have the option to publish the peer review history of their article (what does this mean?). If published, this will include your full peer review and any attached files.

**Do you want your identity to be public for this peer review?** For information about this choice, including consent withdrawal, please see our Privacy Policy.

Reviewer #1: **Yes: **Derbew Fikadu Berhe

Reviewer #2: No

---

## [Editor Report · Decision Letter 2]

6 Dec 2023

Time to Elective Surgery and Its Predictors after First Cancellation at Debremarkos Comprehensive Specialized Hospital, Northwest Ethiopia

PGPH-D-23-00985R2

Dear Dr Zeleke,

We are pleased to inform you that your manuscript 'Time to Elective Surgery and Its Predictors after First Cancellation at Debremarkos Comprehensive Specialized Hospital, Northwest Ethiopia' has been provisionally accepted for publication in PLOS Global Public Health.

Best regards,

Barnabas Tobi Alayande

Academic Editor

Reviewer Comments (if any, and for reference):

The pending concerns of the reviewers have been addressed. Thank you for addressing these directly and in detail.

1. The comment "Data collectors possibly included cases with complete data, which was taken as the main drawback of the study." should be included in the text to adequately explain the statement " Selection bias has been identified to be the primary drawback of this study" (Discussion Paragraph 5, Line 4) This will address the concern of a previous reviewer within the text, as it has been written within the response to reviewer.

2.  Please edit the Ethical Consideration and Consent to Participate appropriately. It reads, "By not keeping the names of the patients anonymous, the confidentiality and privacy of the information were protected." please remove the "not" so that it reads "By keeping the names of the patients anonymous, the confidentiality and privacy of the information were protected" which is what is communicated by the text.